# Experimental Investigation and Modeling of Damage Accumulation of EN-AW 2024 Aluminum Alloy under Creep Condition at Elevated Temperature

**DOI:** 10.3390/ma14020404

**Published:** 2021-01-15

**Authors:** Adam Tomczyk, Andrzej Seweryn

**Affiliations:** Faculty of Mechanical Engineering, Białystok University of Technology, Wiejska 45C Str., 15-351 Białystok, Poland; a.seweryn@pb.edu.pl

**Keywords:** creep-rupture test, damage accumulation, aluminum alloy, SEM observations, theoretical model

## Abstract

The paper is focused on creep-rupture tests of samples made of the 2024 alloy in the T3511 temper under uniaxial tensile stress conditions. The basic characteristics of the material at the temperatures of 100, 200 and 300 °C were determined, such as the Young’s modulus *E*, yield point σ_y_, ultimate tensile strength σ_c_ and parameters *K* and *n* of the Ramberg–Osgood equation. Creep tests were performed for several different levels of nominal axial stress (load) at each temperature. It was observed that in the process of creep to failure at 200 and 300 °C, as the stress decreases, the creep time increases and, at the same time, the strain at rupture increases. However, such a regularity is maintained until a certain transition stress value σ_t_ is reached. Reducing the stress below this value results in a decreased value of the strain at rupture. A simple model of creep damage accumulation was proposed for the stress range above the transient value. In this model, the increase in the isotropic damage state variable was made dependent on the value of axial stress and the increase in plastic axial strain. Using the results of experimental creep-rupture tests and the failure condition, the parameters of the proposed model were determined. The surface of fractures obtained in the creep tests with the use of SEM technology was also analyzed.

## 1. Introduction

One of the earliest and best-known equations describing the creep phenomenon under uniaxial loading (tension) is the Norton power law [1]:(1)$$\frac{d\varepsilon }{dt}=B\sigma^{n}$$
where *B* and *n* are material parameters; ε and σ are the axial strain and stress, respectively; and *t* is time. It allows describing a stable, secondary creep. Nowadays, however, the increasing demands placed on advanced structures and their components subjected to long-term permanent loads over time render this law an insufficient modeling tool. It is necessary to be able to describe all three creep stages. Moreover, it is not enough to know the increase rate but also to be able to predict the accumulation of damage caused by the creep. For modeling the damage increase, a certain scalar parameter (damage state variable), denoted by *D* (e.g., [2,3]) or ω (e.g., [4,5]), is usually introduced, which equals zero for the undamaged material condition and reaches the value of unity for the complete material failure.

One of the first models to describe the damage state was proposed by Kachanov [6] and Rabotnov [7] and was based on the laws of continuum damage mechanics:(2)$$\begin{array}{cc} \frac{d\varepsilon }{dt}=B{\left(\frac{\sigma }{1-\omega }\right)}^{n}, & \frac{d\omega }{dt}=C{\left(\frac{\sigma }{1-\omega }\right)}^{m} \end{array}$$
where ω is the damage parameter mentioned, and *B*, *C*, *n* and *m* are the material constants, with *B* and *n* being related to Norton’s law. Currently, we can find many modifications of this model consisting of the generalization of multiaxis states by introducing reduced values of stress σ_eq_ and strain ε_eq_. The values of hydrostatic pressure and principal stresses (e.g., [8,9]) or the values of tensor invariants and the stress deviator (e.g., [10]) are often used here. Rapid elastic–plastic damage dominates in the tertiary creep rather than classical creep damage, which is not captured well enough by the type (2) equations. In order to eliminate this inconvenience, a model using the exponential function was proposed by Liu and Murakami [11]. It was successfully used in predicting creep crack growth [12]. In the paper by Othman et al. [13], the damage accumulation measure was treated as the composition of two: ω_1_ and ω_2_. One of them (ω_1_) is related to the dislocation density, and the other (ω_2_) is defined as the area fraction of cavitation under the creep constraint and reaches a value of 1/3 at rupture. This model was also used in a generalized form for complex stress states [14]. There are more models in which the damage state variable is considered the sum of the two components. To take into account material softening in the tertiary creep due to void nucleation at the grain boundary and to precisely describe the primary creep stage in aluminum alloys, Kowalewski et al. [15] proposed to introduce additional parameters *h* and *H*, which relate to the primary creep stage and represent material hardening. Value *H* peaks at the end of the primary creep and remains constant in the tertiary. The generalization of this model into complex stress states can be found in many papers (e.g., [16,17]). Parameter *H* was later used many times in modeling practical engineering creep problems (e.g., [18,19,20,21,22]).

To predict the crack propagation in [23,24] the model describing an increase in damage in the zone in front of the crack tip is presented:(3)$$\frac{d\omega }{dt}=\frac{\dot{\varepsilon }}{\varepsilon_{f}^{*}}$$
where $\varepsilon_{f}^{*}$ is the strain at rupture in the complex load state, and $\dot{\varepsilon }$ is the creep rate. Parameter $\varepsilon_{f}^{*}$ takes into account the multiaxial effect [25], namely:(4)$$\frac{\varepsilon_{f}^{*}}{\varepsilon_{f}}=\frac{\sinh\left(\frac{2}{3}\frac{n-0.5}{n+0.5}\right)}{\sinh\left(2\frac{n-0.5}{n+0.5}\frac{\sigma_{h}}{\sigma_{\mathrm{eq}}}\right)}$$
where σ_h_ and σ_eq_ are hydrostatic and reduced stresses, respectively, according to Huber–von Mises; ε_f_ is the strain at fracture in the uniaxial state; and *n* is the stress exponent in the secondary creep. This model allowed for analyzing the crack initiation and propagation for P91 steel, Cr-Mo-V steel, ceramic materials or P92 steel [26,27,28,29]. In order to better predict the increased rate of void growth, Wen and Tu [30] proposed replacing the “sinh” function in (4) with an exponential function. Other formulas taking into account the multiaxiality factor $\varepsilon_{f}^{*}$ can also be found in the literature [31,32,33].

The observation and registration (using an optical digital microscope) of density changes *D*_c_ of the crazing on the surface subjected to the creep process enabled the formulation of the relationship [5]:(5)$$\omega =\frac{\mathrm{lg}\left(t\right)}{\mathrm{lg}\left(t_{r}\right)}$$
where *t* and *t*_r_ denote the current creep time and the time to rupture, respectively. An even simpler form of the damage state variable was used in the paper of Cardoso et al. [34]:(6)$$D=\frac{t}{t_{r}}$$

In this case, the results of hardness measurements were used to determine the material damage. The use of a parameter similar to the parameter provided above was also proposed by Rui et al. [35], taking into account the results of the EBSD analysis of austenitic steels. The so-called grain reference orientation deviation parameter (*GROD*) was introduced into the model. It was demonstrated that the relationship between the creep strain and the *GROD* parameter is linear and allows easy modeling of damage growth.

Structural elements are often subject to “cyclic” creep behavior. At time *t*_i_ and temperature *T*_i_, some stress σ_i_ is subjected. Robinson’s rule is then used to sum the creep damage [36] derived from the Palmgren–Miner fatigue hypothesis:(7)$$D=\sum_{i=1}^{n}\frac{t_{i}}{t_{\mathrm{fi}}}$$
where *t*_fi_ denotes the time to failure under defined conditions. This model was used by Borkowski et al. [37] to determine damage to steam turbine blades and by Loghman and Moradi [38] to analyze creep effects in a thick-walled spherical reactor. The work of Hu et al. [39] shows the strain form of Relationship (7) by replacing the time values with the corresponding strains.

Because Robinson’s linear rule did not reflect the actual experimental results well enough, Pavlou [40] proposed to determine damage using the graphical dependence of the stress logarithm on the Larson–Miller parameter. To consider load history for the cycle block during creep, Batsoulas [41] introduced the function of cumulative creep damage in the form:(8)$$\frac{d\omega }{\omega }=\phi \left(t,\sigma ,T\right)\frac{d\varepsilon }{\varepsilon }$$
where *ϕ*(*t*, σ, *T*) is a function of time, stress and temperature. The integration of Equation (8) will allow obtaining the current value of the damage. Hu et al. [39], pointing to the shortcomings of the approach presented in [40,41], propose the use of the creep damage tolerance parameter *λ*_f_ dependent on the minimum strain rate. At the same time, both the approach proposed by Hu et al. and the Pavlou model for the titanium alloy were verified.

Wen et al. [42] presented the total damage *D* of a nickel-based single crystal as the damage *D*_S_ resulting from microstructural material degradation and cavitation damage ω. To describe the increase rate ω, the nondilatational strain rate was used. Pettinà et al. [43] presented total damage as the damage *D*^cr^, resulting clearly from creep effects and damage *D*^env^ related to the oxidizing influence of the environment. This allowed taking into account the process of creep oxidation of ZrN ceramics at high temperatures.

The paper of Dyson [44] takes into account several micromechanisms related to damage accumulation during creep. The effects of parameters such as depletion of the Laves phase, cavity density in the grain boundary, precipitation particle coarsening and dislocation damage are described in detail.

In the above-mentioned relationships describing the damage growth, the scalar variable of the damage state (*D* or ω) is used. In fact, this variable can be treated as a special case of the general, tensor damage measure, similarly to isotropic damage models, which can be treated as a special anisotropic case. Indeed, the creep process is, in its nature, anisotropic. To describe it in the creep process, Chaboche [45] used a fourth-order damage tensor **D** by introducing an effective stress tensor. Murakami [46,47] used a second-rank tensor to describe the growth and joining of voids at grain boundaries in a representative volume. Ganczarski and Skrzypek [48] pointed out the fact that the model proposed by Murakami did not allow for sufficiently accurate modeling of damage growth in axially symmetric cases, especially in the area close to the axis of symmetry, and presented its modification. A more detailed review of the anisotropic damage models used to describe the creep process and their applications can be found in [49,50]. The damage accumulation model in the creep weld material is proposed by Peravali et al. [51]. It was assumed that the damage increases in the plane of maximum normal stresses both along and across the welding direction. A second-order damage tensor was used here.

This study concerns short-term creep-rupture tests for aluminum alloy EN-AW 2024 (T3511), which were carried out at the temperatures of 100, 200 and 300 °C. Depending on the load, the creep time ranged from several dozen minutes to one hundred and several dozen hours. A microscopic analysis of fractures was also performed, thanks to which the mechanisms of material damage and fracture were identified. Based on the test results and the analysis of the failure mechanism, a simple, two-parameter model of damage accumulation is presented. This, together with experimental verification, is the main aim of the paper. An increase in the damage state variable *ω* was made dependent on an increase in plastic strain and the value of the defined stress. The parameters of the model were determined, and the possibilities of its application were indicated. The proposed model has never been used in modeling creep problems before. Moreover, the influence of different creep stress and creep times on the nature of fractures and the failure mechanism are presented.

## 2. Test Stand, Material Characteristic, Samples

The tested material was EN-AW 2024 aluminum alloy in the T3511 temper, in the form of extruded bars with a diameter of 16 mm. The shape and dimensions of the samples used in the monotonic tensile tests and creep-rupture tests are shown in Figure 1. The chemical composition (weight %) of the tested alloy is [52]: Si (0.13%), Fe (0.25%), Cu (4.4%), Mn (0.62%), Mg (1.7%), Cr (0.01%), Zn (0.08%), Ti (0.05%) and Al (bal.).

The view of the microstructure of the material in the longitudinal and transversal cross-sections can be found in an earlier paper [52]. The samples were made by turning with the use of a numerically controlled lathe. This guaranteed the repeatability of dimensions as well as an identical load pattern for each sample in the machining process. The threaded gripping parts of the specimen ensured secure clamping and eliminated the possibility of the specimen slipping out of the holder due to elevated temperature.

Tensile and creep-rupture tests at an elevated temperature were carried out using a Zwick/Roell Kappa 100SS four-column creep-testing machine (Zwick/Roell, Ulm, Germany) with an electromechanical drive (Figure 2a). It allows for the performance of creep, creep-rupture and stress relaxation tests at loads up to 100 kN. The sample was mounted in the loading system by means of high-temperature tension strings. The articulated end of one of them guaranteed that the load had the form of pure tension. The thermal load was generated by a Maytec three-zone furnace (Mess und Regeltechnik GmbH, Singen, Germany) with an upper temperature range of 900 °C, controlled by a special controller. Three thermocouples allowed for precise measurement of the sample temperature. Due to the small measurement base, only two thermocouples were used, i.e., the upper and lower. The TestXpertII system controlled both the creep-testing machine and the furnace. Before starting the actual test, the sample was heated at the rate suggested by the furnace manufacturer, i.e., 3 °C/min for the test temperature of 100 °C, 6 °C/min for 200 °C and 10 °C/min for 300 °C. Each of the temperatures was achieved with an accuracy of ±2 °C. The temperature of 100 °C was reached after about 100 min, 200 °C after approx. 85 min and 300 °C after 65 min. The heating time tolerance was ±5 min. The strain of the gauge length of the sample inside the furnace was measured with the use of a special device [53] (Figure 2b). It allows the measurement of sample elongations at an elevated temperature outside the furnace using devices designed for operation at room temperature. The Epsilon 3542-025M-025-HT1 extensometer (Epsilon Technology Corp, Jackson, MS, USA) with a gauge length of 25 mm and a range of ±12.5 mm was used as a precise measuring device. The extensometer was calibrated before each single test. In both the monotonic tensile and creep-rupture tests, the samples were loaded with the same strain rate of 0.0015/s. After the end of the test, the samples were cooled always under the same conditions, i.e., in the open air.

The process of modeling damage accumulation in the creep process described in the following sections requires knowledge of the selected monotonic characteristics of the material. These include Young’s modulus *E* and tensile strength σ_c_ as well as parameters *K* and *n* of the Ramberg–Osgood equation [54]:(9)$$\varepsilon =\varepsilon_{e}+\varepsilon_{p}=\frac{\sigma }{E}+{\left(\frac{\sigma }{K}\right)}^{\frac{1}{n}}$$
where σ and ε are the stress and the corresponding strain, and ε_e_ and ε_p_ are the elastic and plastic strains. These material characteristics were determined based on the monotonic tensile tests at 100, 200 and 300 °C [52,55,56] and are summarized in Table 1. It should be emphasized that the monotonic tensile tests were carried out for three samples at each temperature, and their results were averaged. Note that when setting the parameters *n* and *K*, only the range of the stress–strain curve was taken into account until the neck was formed in the sample, i.e., until the ultimate tensile strength value σ_c_ was reached. These parameters were obtained by approximating the results of the experimental tests obtained in the monotonic tensile test at different temperatures by Equation (9). The diagrams of monotonic tensile tests at 100, 200 and 300 °C obtained in the experiment and by approximation are shown in Figure 3. The approximation was also limited to the strain value corresponding to stress σ_c_.

## 3. Experimental Results and Discussion

### 3.1. Creep-Rupture Tests

The creep tests were performed at temperatures of 100, 200, 300 °C, according to the ISO standard [57] at five different constant force values at temperatures of 200 and 300 °C and at three values at 100 °C (Table 2). The creep process was repeated three times with the same load value, and the results were averaged.

Photos of typical ruptured samples after the creep process are shown in Figure 4. The curves of the nominal axial strain versus time at each temperature for different load levels are shown in Figure 5.

Due to significantly different creep times for different values of the nominal stress value σ_creep_ at the same temperature, the diagrams of strain versus time are decomposed into two with different time scales. As can be seen, the creep process of the tested alloy has a classic, three-stage character, like most aluminum alloys (e.g., [58,59,60]). Under creep conditions at 200 and 300 °C, as the nominal stress (load) decreases, the creep time and strain at rupture increase. However, such a tendency can be observed only within a certain load range, corresponding to the time range from zero to approximately twenty hours. After the load is reduced below a certain transient value of the nominal stress σ_t_, which has a certain limit value for the creep time *t*_t_, the tendency is reversed. There is a clear decrease in the strain at rupture with a decrease in load, which is, of course, accompanied by an increase in the creep-rupture time.

This is related to the evolution of the material microstructure as a result of long-term thermal and mechanical loading and may also apply to other aluminum alloys (e.g., [61]). The results of the single tests in which this tendency was observed are highlighted in gray in Table 2. However, to precisely determine the aforementioned transient load and time values, additional creep-rupture tests should be performed.

The creep process can also be represented schematically in the nominal stress–strain (σ–ε) system (Figure 6). Section OA corresponds to the elastic range. Section AB describes the behavior of the material after reaching the yield point σ_y_ until the moment when a defined constant force in the creep process is reached. What corresponds to this moment is a certain plastic strain ε_p1_. The Ramberg–Osgood (R–O) equation (Equation (9)) was used to describe the AB section. Section BC corresponds to the creep process at constant force until specimen rupture under permanent strain ε_p2_. Note that σ_creep_ is the nominal stress determined for the constant cross-section of the specimen. Strain values ε_p1_ and ε_p2_ can be found in Table 2. In the creep-rupture process at 100 °C, the values of nominal stress above the yield point of the material at this temperature were used. In the creep-rupture tests at temperatures of 200 and 300 °C, the nominal stress values under the yield point of the material at these temperatures were used. Thus, the deformation value ε_p1_ was equal to zero (Figure 6).

### 3.2. Microscopic Analysis of Fracture Surfaces

Surface tests of fractures obtained in the creep-rupture tests were carried out using the Olympus SXD110 optical microscope (Olympus Corp, Essex, UK) and the Phenom XL scanning electron microscope (SEM) (Thermo Fisher Scientific, Waltham, MA, USA). The first of the devices allows for magnification from 20 to 1071 times. The Phenom microscope is fitted with a backscattered electron detector (BSD), a secondary scattered electron detector (SED) and an energy-dispersion spectrometer (EDS). It allows for magnifications from 80 to 100,000 times.

Creep to failure of the investigated alloy at 100 °C gives a fracture surface (Figure 7) similar to the surface obtained in the case of monotonic tensile (e.g., [52]). The orientation angle of the fracture plane (approx. 45°) proves the dominant share of maximum shear stresses in the fracture process. However, numerical calculations show that the maximum value is reached by principal stresses σ_1_ in the symmetry axis [52]. In the central area, normal stress initially dominates, as evidenced by the SEM images shown in Figure 7c,d. At the bottom of almost each of the pores, coarse precipitates can be clearly seen. The fracture process began in the sample axis. Due to the effect of stress σ_1_, the pores elongated, and thus their diameter decreased. As soon as the wall (bridge between the pores) touched the sharp edge of the precipitate, it was ruptured, which caused two adjacent pores to join. The process of normal stress domination lasted for a relatively short time, as evidenced by the slightly deformed remains of ruptured bridges. The crack was initiated in the sample axis after which rapid shearing occurred. At high magnifications, it can be noticed that as the values of σ_creep_ decrease, and thus the creep time increases, the diameter of the cavities increases (Figure 7c,d). At the same time, there was a growing tendency for smaller pores to join into larger ones. The rupturing of the bridges happened before the pores could join.

The fracture surface obtained in the creep-rupture tests at 200 °C has a two-plane character (Figure 8a–c). The central part is occupied by a plane perpendicular to the load direction (central region). Its smoothness increases with an increase in the stress value σ_creep_. The other part of the characteristic planes (external region) connects the central region with the external surface of the sample and is oriented at an angle of approximately 45° relative to the load direction. The share of the first of these regions in the entire cross-section decreases as the load increases. The border between the two areas is very clear, as shown in Figure 9f. The failure mechanism thus becomes less and less ductile. The pores within the central surface were clearly deformed in the direction of the principal stress *σ*_1_ (Figure 8d,f). Numerical analysis of the monotonic tensile process (e.g., [52]) shows that as soon as the neck forms, the stress state becomes triaxial. However, failure is determined not by the maximum equivalent stresses σ_eq_ but by the maximum principal stresses σ_1_. Fracture starts in the sample axis similarly to creep at 100 °C. Here, however, the dominance of the principal stresses lasts much longer. The crack propagates perpendicular toward the load direction for a longer time. With longer times of exposure to elevated temperature, the cavities have larger diameters (see Figure 8d,f). At the same time, the pores tend to join into larger ones. There are not many of the finest pores shown in Figure 8d. The character of the external region in both cases is very similar (Figure 8e); hence, for the stress σ_creep_ = 297 MPa, there was no need to present its view. Here, it was the maximum shear stress that was responsible for the failure. A large number of fine pores that have not grown significantly but have been sheared rapidly can be seen. There are also a few remnants of larger pores that were initially deformed in the direction of stress σ_1_, as evidenced by the presence of stripes on the surfaces of the bridges. Subsequently, the bridges between the pores were sheared rapidly.

The surface of fractures obtained in the creep-rupture tests at 300 °C had two planes (central and external regions) only for high loads and thus for short creep times (Figure 9c). The share of the shear surface area (external region) here is small. In the case of lower stress values, the character of the fracture was clearly ductile, and the entire area was dominated by stress σ_1_.

At the same time, the fracture surfaces of both parts of the same sample following the rupture were not “negative” and “positive,” as was in all other cases. Due to the prolonged exposure to elevated temperature, the bridges between the pores in the central region were broken. The crack initiating in the axis of symmetry propagated toward the outer surface. During this time, the material on the outside was still slowly deforming in the direction of the loading. The final result is shown in Figure 9a,b. The diameter of the cavities after ruptured pores increased as the stress σ_creep_ decreased. Such a situation was observed for σ_creep_ = 51 MPa (Figure 9d) compared to σ_creep_ = 75 MPa (not shown in the figure). However, even at higher loads, such as σ_creep_ = 123 MPa, the central region was characterized by larger and more regular shapes of the cavities (Figure 9e). This is due to the fact that the pores were not significantly deformed in the direction of the stress *σ*_1_ action and were ruptured earlier. In the external area, there are cavities after the pores sheared in the direction of maximum shear stress acting. The presence of numerous stripes on the surface of the cavities (Figure 9f) indicates that the shearing process at 300 °C was not as rapid as at 200 °C.

## 4. A Simple Model for Creep Damage Accumulation

In the proposed computational model, it was assumed that in the creep-rupture process, damage growth depends on the value of the axial stress and an increase in permanent (plastic) strain. By introducing a measure of damage state ω, the law of damage was adopted in the form used to predict the fatigue life of sintered porous steels [62]:(10)$$d\omega =\{\begin{array}{cc} A_{\omega }{\left(\frac{\sigma }{\sigma_{c}}\right)}^{n_{\omega }}{d\varepsilon }_{p} & \mathrm{dla}\;\sigma >0\;i\;{d\varepsilon }_{p}>0\\ 0 & \mathrm{dla}\;\sigma \leq 0\;\mathrm{lub}\;{d\varepsilon }_{p}\leq 0 \end{array}$$
where: *A*_ω_, *n*_ω_—material parameters independent of the loading value; σ—the current value of the tensile stress; σ_c_—nominal critical stress for the undamaged material (the ultimate tensile strength was assumed).

The failure criterion was adopted in a simple form:(11)ω=1

Value ω = 0 means undamaged material. The total damage growth Δω during the creep-rupture process can be expressed as the sum of two components:(12)$$\Delta \omega =\Delta \omega_{1}+\Delta \omega_{2}$$
where: Δω_1_, Δω_2_—growth of the ω, respectively, in sections *AB* and *BC* (Figure 6) and according to Equation (10):(13)$$\Delta \omega_{1}=\int_{0}^{\omega_{1k}}{d\omega }_{1}=\frac{A_{\omega }}{\sigma_{c}^{n_{\omega }}}\int_{0}^{\varepsilon_{p1}}\sigma^{n_{\omega }}{d\varepsilon }_{p}$$
(14)$$\Delta \omega_{2}=\int_{\omega_{1k}}^{\omega_{2k}}{d\omega }_{2}=A_{\omega }{\left(\frac{\sigma_{\mathrm{creep}}}{\sigma_{c}}\right)}^{n_{\omega }}\int_{\varepsilon_{p1}}^{\varepsilon_{p2}}{d\varepsilon }_{p}$$
where: σ_creep_—nominal (engineering) stress, corresponding to the constant force in the creep process; ε_p1_—permanent strain, corresponding to the beginning of the stage of reaching the required force in the creep process (Point B, Figure 4); ε_p2_—permanent strain, corresponding to sample rupture (Point C, Figure 4); ω_1k_, ω_2k_—values of the damage state variable corresponding to deformations ε_p1_ and ε_p2_.

Assuming the hardening curve in the form consistent with Equation (9), i.e.,:(15)$$\sigma =K{\left(\varepsilon_{p}\right)}^{n}$$

Equations (13) and (14) will finally take the following form:(16)$$\Delta \omega_{1}=A_{\omega }{\left(\frac{K}{\sigma_{c}}\right)}^{n_{\omega }}\int_{0}^{\varepsilon_{p1}}{\left(\varepsilon_{p}\right)}^{{\mathrm{nn}}_{\omega }}{d\varepsilon }_{p}={\left(\frac{K}{\sigma_{c}}\right)}^{n_{\omega }}\frac{A_{\omega }}{\left(nn_{\omega }+1\right)}{\left(\varepsilon_{p1}\right)}^{\left(nn_{\omega }+1\right)}$$
(17)$$\Delta \omega_{2}=A_{\omega }{\left(\frac{\sigma_{\mathrm{creep}}}{\sigma_{c}}\right)}^{n_{\omega }}\left(\varepsilon_{p2}-\varepsilon_{p1}\right)$$
(18)$$\Delta \omega \left(\varepsilon_{p}\right)=\{\begin{array}{cc} {\left(\frac{K}{\sigma_{c}}\right)}^{n_{\omega }}\frac{A_{\omega }}{\left(nn_{\omega }+1\right)}{\left(\varepsilon_{p}\right)}^{\left(nn_{\omega }+1\right)} & \mathrm{for}\;\varepsilon_{p}\leq \varepsilon_{p1}\\ {\left(\frac{K}{\sigma_{c}}\right)}^{n_{\omega }}\frac{A_{\omega }}{\left(nn_{\omega }+1\right)}{\left(\varepsilon_{p1}\right)}^{\left(nn_{\omega }+1\right)}+A_{\omega }{\left(\frac{\sigma_{\mathrm{creep}}}{\sigma_{c}}\right)}^{n_{\omega }}\left(\varepsilon_{p}-\varepsilon_{p1}\right) & \mathrm{for}\;\varepsilon_{p1}<\varepsilon_{p}\leq \varepsilon_{p2} \end{array}$$

Adopting the failure condition in the form (11), we obtain:(19)$$\omega =\Delta \omega_{1}+\Delta \omega_{2}={\left(\frac{K}{\sigma_{c}}\right)}^{n_{\omega }}\frac{A_{\omega }}{\left(nn_{\omega }+1\right)}{\left(\varepsilon_{p1}\right)}^{\left(nn_{\omega }+1\right)}+A_{\omega }{\left(\frac{\sigma_{\mathrm{creep}}}{\sigma_{c}}\right)}^{n_{\omega }}\left(\varepsilon_{p2}-\varepsilon_{p1}\right)=1$$

When ε_p1_ = 0:(20)$$\omega =\Delta \omega_{2}=A_{\omega }{\left(\frac{\sigma_{\mathrm{creep}}}{\sigma_{c}}\right)}^{n_{\omega }}\varepsilon_{p2}=1$$

The values of *A*_ω_ and *n*_ω_ obtained using Conditions (19) or (20) are summarized in Table 3. The values of the damage state variable ω obtained for the mentioned parameters in the creep-rupture process at temperatures of 100, 200 and 300 °C are also presented here. The simple gradient method and the systematic search method were used in numerical algorithms.

It should be emphasized that in the procedure of determining parameters *A*_ω_ and *n*_ω_, the results of the creep tests, in which a decrease in the value of the strain at rupture was observed with a decrease in the nominal stress, were omitted. This means that the proposed model is not applicable to this load range. They are highlighted in gray in Table 3. However, for these cases also, the values of Δω_2_ and ω were calculated and listed in the table. As can be seen, these values are very far from unity. Using the determined parameters *A*_ω_ and *n*_ω_ and the damage growth law in (18), the value of the damage state variable was determined for different values of the load at a given temperature as a function of the current plastic strain ε_p_ (Figure 10). On the other hand, Table 3 presents the values of the damage state variable ω determined by the computational model at the moment of the actual (determined experimentally) sample failure. They are in the range from 0.954 to 1.035 (except the gray cells), which means that the sample failure prediction error did not exceed 4.6%. At the same time, it can be seen how much the lines corresponding to σ_creep_ = 229 MPa and σ_creep_ = 235 MPa in Figure 10b and σ_creep_ = 51 MPa in Figure 10c do not satisfy the condition ω = 1.

## 5. Conclusions

This paper presents the results of experimental creep-rupture tests of aluminum alloy EN-AW 2024 T3511 at temperatures of 100, 200 and 300 °C under uniaxial stress conditions. The results prove that the fracture process always begins in the sample axis as a result of the action of maximum principal stresses σ_1_. Pores are deformed in the direction of the loading action, and the initiator of the bridge rupturing is usually a coarse precipitate. Adjacent pores join as a result of the rupturing of successive bridges, enlarging the crack. It always propagates from the axis toward the outer sample surface. In the case of lower temperatures and higher load values, the time of normal stress domination is short, and failure occurs by rapid shearing of the material. Due to higher temperature values and lower loads, the failure is primarily determined by normal stress.

The results of the experimental tests were the basis of the proposed simple model, allowing to determine the damage state, in particular the moment of failure of samples caused by the creep at an elevated temperature. In this model, an increase in the damage state variable dω was made dependent on the increase in plastic strain and the value of stress loading the sample. It was assumed that the damage state variable ω is the sum of two components ω_1_ and ω_2_. The first term (ω_1_) describes the accumulation of creep damage for stresses in the range above the yield point until a constant force is established. The second term (ω_2_) defines the method of adding damage from the moment of reaching the value of the constant load to material failure. In special cases, i.e., when the creep load does not exceed the yield point of the material, the first of the said terms is zero. The proposed model requires the determination of only two material parameters, which is possible based on the analysis of the results of the experimental creep-rupture tests at a specific temperature. The advantage of these parameters is their independence from the loading value, but only from the temperature, which was demonstrated in the paper. A method of determining these parameters was also proposed, and their values for three different temperatures were presented. The scope of the model’s applicability was indicated.

Using the strain–creep time dependencies known from the literature, the damage state variable can be represented as a function of time. It will allow, for example, the determination of the time to failure under given loading and temperature conditions.

The results of the experimental tests made it possible to indicate the presence of a certain transition value of the creep stress σ_t_, a corresponding creep time value *t*_t_ and strain at rupture ε_t_. Until these values are reached, the creep time increases with a decrease in the load value, and the strain value at the time of failure increases simultaneously. After exceeding these transient values, the decrease in load is accompanied by an increase in the creep failure time, with a simultaneous decrease in the value of the strain at rupture. This was the case for 200 and 300 °C, where the loading did not exceed the yield point of the material at a specific temperature. This is a consequence of the loaded material residing at an elevated temperature for a sufficiently long time.

## Figures and Tables

**Figure 1 materials-14-00404-f001:**
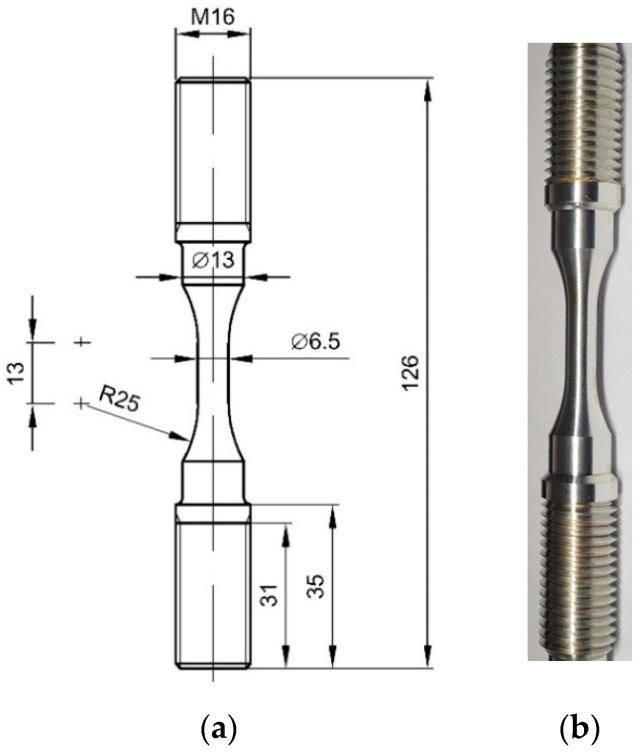
Samples used in the creep-rupture tests: (**a**) dimensions in mm; (**b**) view.

**Figure 2 materials-14-00404-f002:**
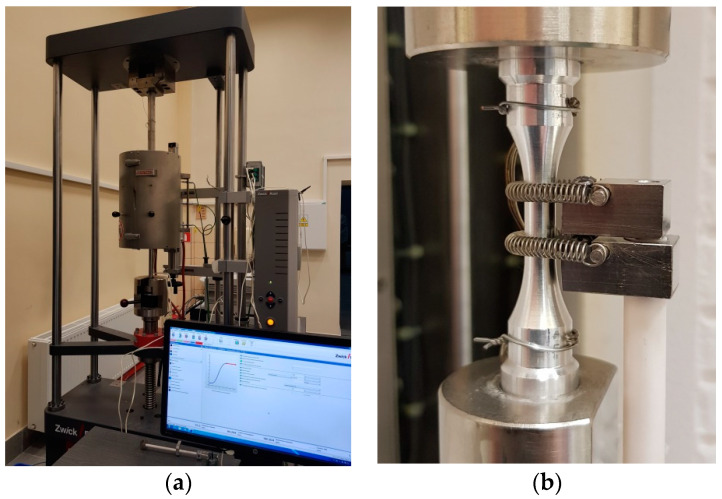
Test stand: (**a**) view of the Kappa 100SS creep-testing machine with an installed Maytec furnace; (**b**) view of the sample in the strings with mounted thermocouples and the special device.

**Figure 3 materials-14-00404-f003:**
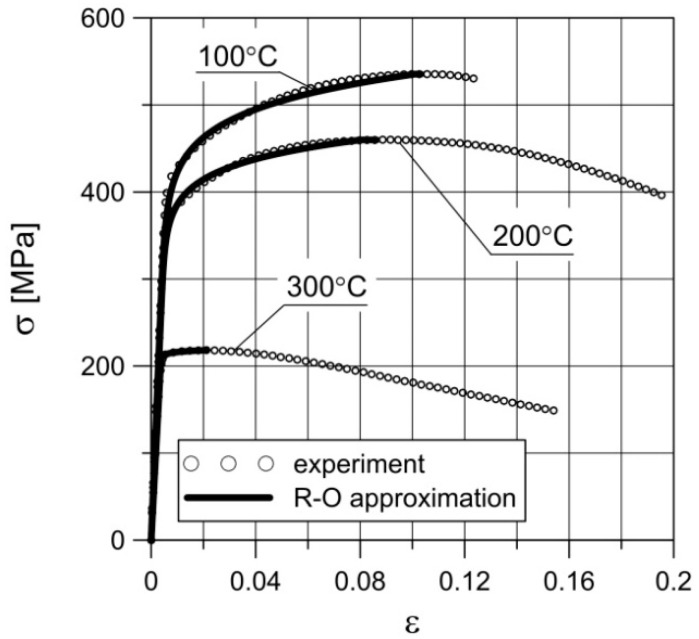
Nominal stress–strain diagrams obtained in the experiment and as a result of approximation by the Ramberg–Osgood (R–O) equation.

**Figure 4 materials-14-00404-f004:**
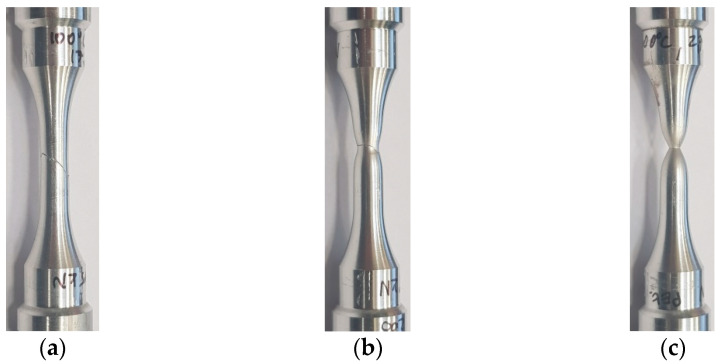
Samples ruptured during creep tests at temperatures: (**a**) 100 °C (σ_creep_ = 532); (**b**) 200 °C (σ_creep_ = 235 MPa); (**c**) 300 °C (σ_creep_ = 75 MPa).

**Figure 5 materials-14-00404-f005:**
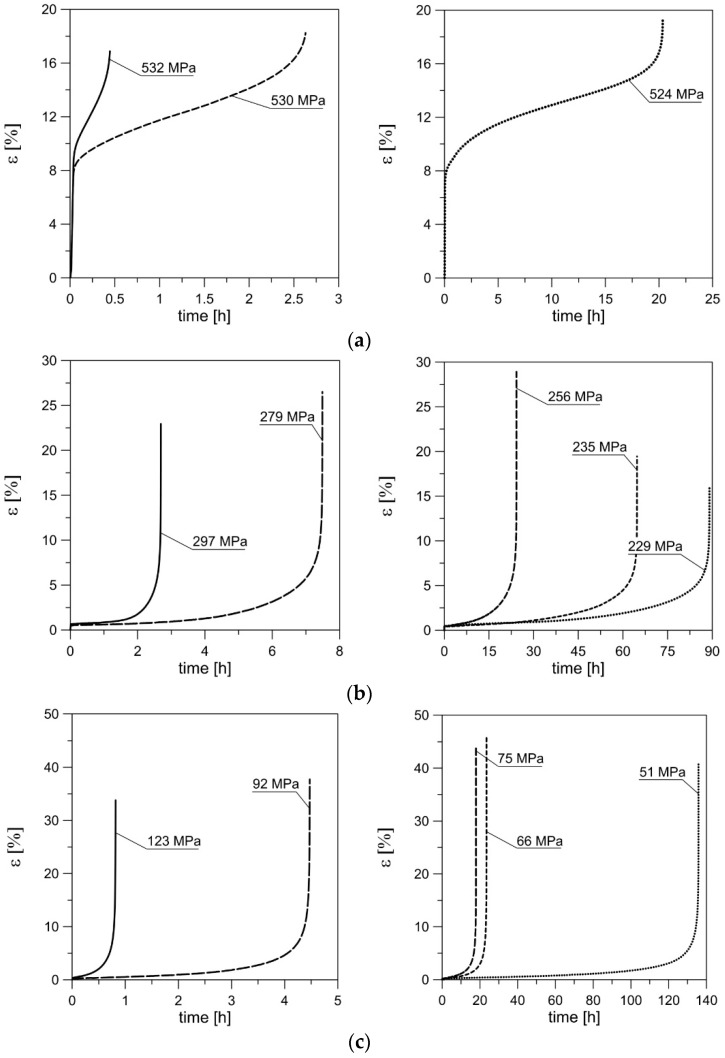
Nominal axial strain vs. time to rupture for different values of nominal stress σ_creep_ at temperatures: (**a**) 100 °C; (**b**) 200 °C; (**c**) 300 °C.

**Figure 6 materials-14-00404-f006:**
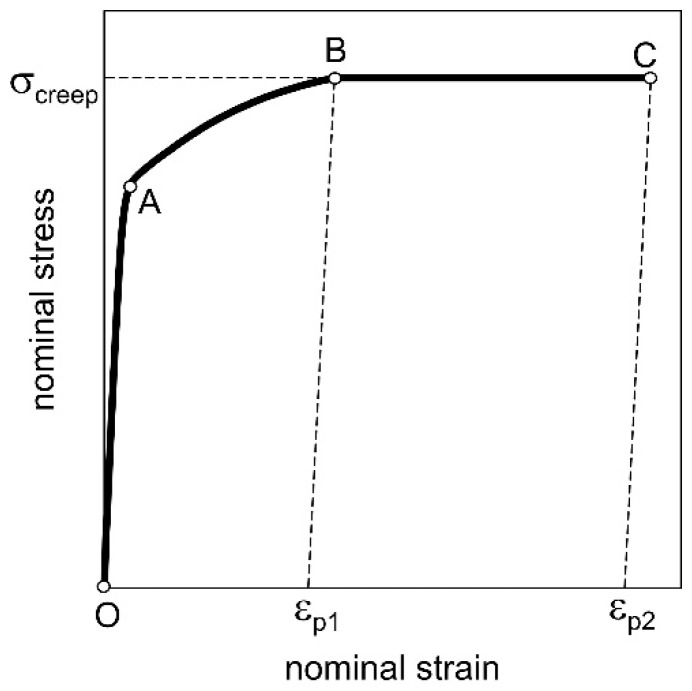
Schematic representation of the creep-rupture process in the nominal stress–strain diagram (σ–ε).

**Figure 7 materials-14-00404-f007:**
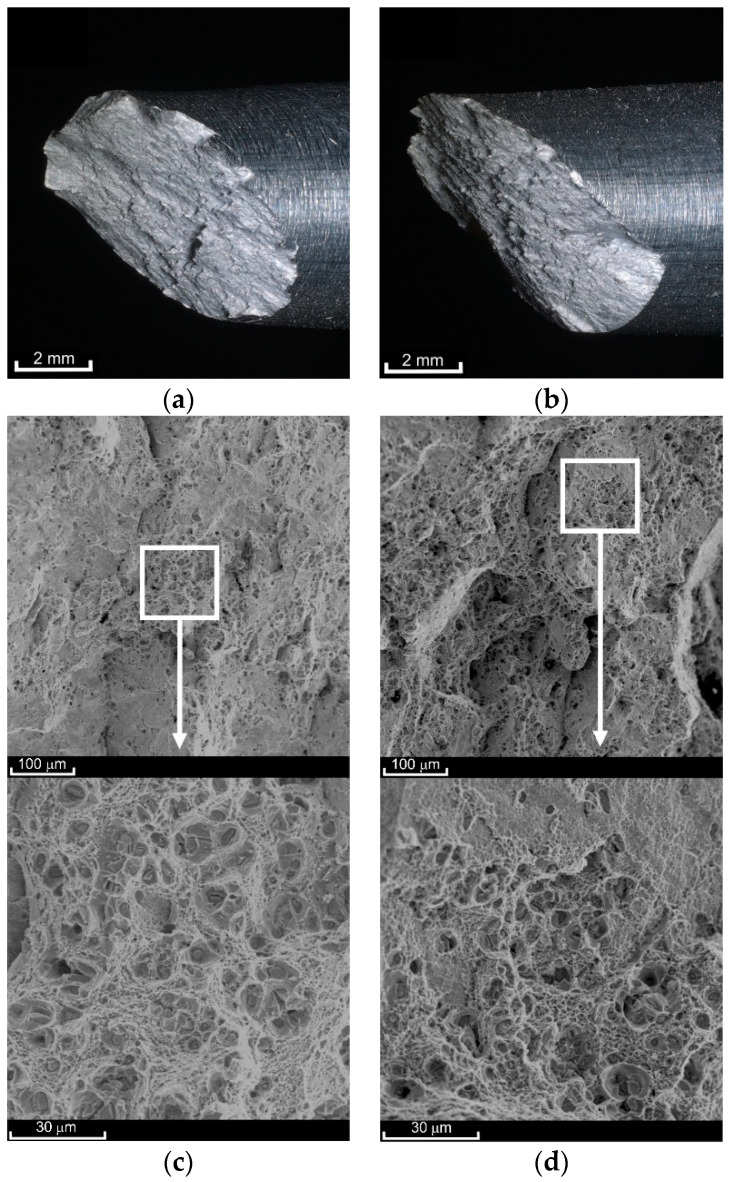
View of the fracture surfaces after creep-rupture tests at a temperature of 100 °C at the macroscale ((**a**) σ_creep_ = 524 MPa; (**b**) σ_creep_ = 532 MPa) and SEM scale ((**c**) σ_creep_ = 524 MPa; (**d**) σ_creep_ = 532 MPa).

**Figure 8 materials-14-00404-f008:**
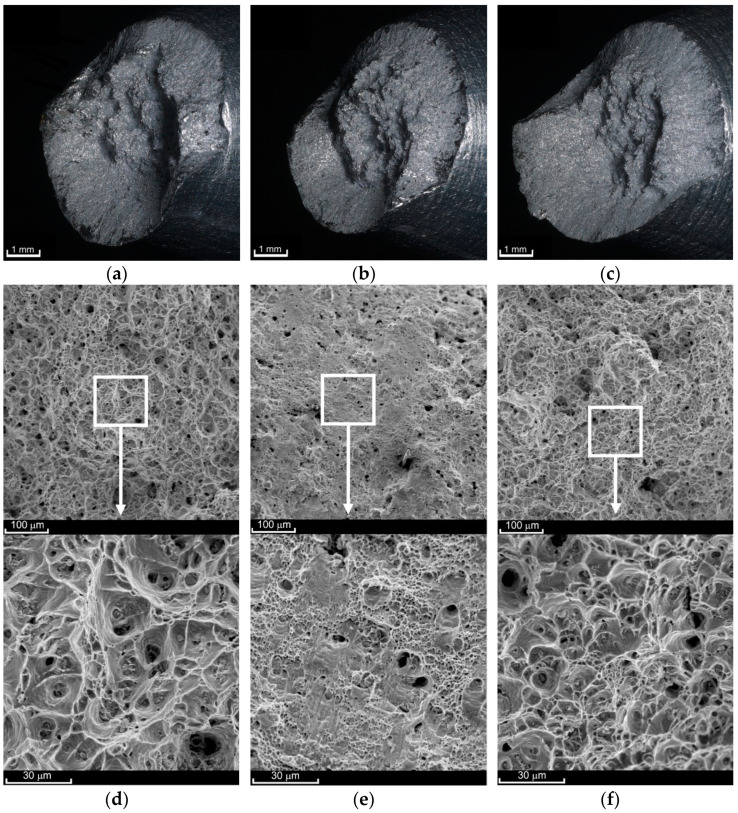
View of fracture surfaces after the creep-rupture tests at a temperature of 200 °C at the macroscale ((**a**) σ_creep_ = 229 MPa; (**b**) σ_creep_ = 256 MPa; (**c**) σ_creep_ = 297 MPa) and SEM scale ((**d**) σ_creep_ = 229 MPa, central region; (**e**) σ_creep_ = 229 MPa, external region; (**f**) σ_creep_ = 297 MPa, central region).

**Figure 9 materials-14-00404-f009:**
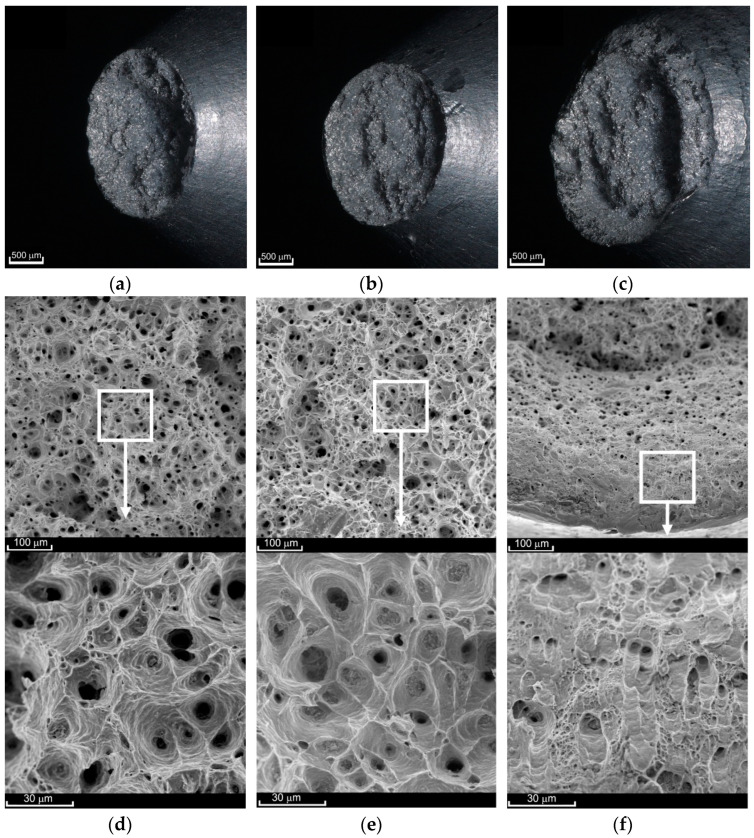
View of fracture surfaces after creep-rupture tests at a temperature of 300 °C at the macroscale ((**a**) σ_creep_ = 51 MPa; (**b**) σ_creep_ = 75 MPa; (**c**) σ_creep_ = 123 MPa) and SEM scale ((**d**) σ_creep_ = 51 MPa; (**e**) σ_creep_ = 123 MPa, central region; (**f**) σ_creep_ = 123 MPa, external region).

**Figure 10 materials-14-00404-f010:**
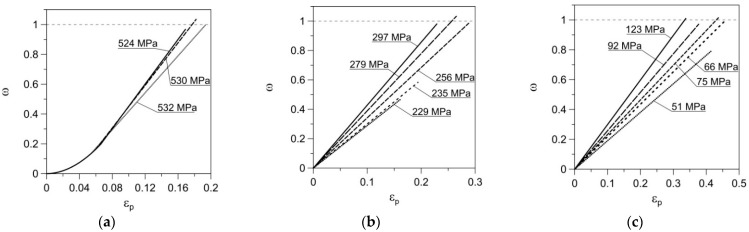
Damage evolution during creep process for different values of nominal stress versus nominal plastic strain at temperatures: (**a**) 100 °C; (**b**) 200 °C; (**c**) 300 °C.

**Table 1 materials-14-00404-t001:** Values of material parameters obtained in the monotonic tensile tests.

*T* (°C)	*E* (GPa)	σ_c_ (MPa)	*n*	*K* (MPa)
100	71	536	0.0759	641
200	70	460	0.0633	542
300	56	219	0.0097	227

**Table 2 materials-14-00404-t002:** Values of time and plastic strain observed in the creep-rupture tests at different temperatures and for different values of σ_creep_.

*T* (°C)	σ_creep_ (MPa)	Creep-Rupture Time (h)	ε_p1_ (%)	ε_p2_ (%)
100	524	20.35	5.4	19.3
	530	2.64	6.0	18.2
	532	0.45	6.6	16.9
200	229	89.05	0	16.2
	235	64.73	0	19.4
	256	24.25	0	28.8
	279	7.49	0	26.5
	297	2.69	0	22.9
300	51	135.83	0	41.7
	66	23.50	0	45.9
	75	17.93	0	43.7
	92	4.47	0	37.7
	123	0.82	0	33.8

**Table 3 materials-14-00404-t003:** Values of *A*_ω_ and *n*_ω_ obtained during the creep process at different temperatures and for different values of σ_creep_ and also the value of the damage parameter ω.

*T* (°C)	σ_creep_ (MPa)	*n* _ω_	*A* _ω_	Δω_1_	Δω_2_	ω
100	524	12.46	8.21	0.134	0.861	0.995
	530			0.164	0.871	1.035
	532			0.198	0.756	0.954
200	229			0	0.468	0.468
	235			0	0.583	0.583
	256	1.51	8.29	0	0.985	0.985
	279			0	1.033	1.033
	297			0	0.981	0.981
300	51			0	0.795	0.795
	66			0	0.998	0.983
	75	0.51	4.01	0	1.015	1.029
	92			0	0.971	0.985
	123			0	1.010	1.005

## Data Availability

The data presented in this study are available on request from the corresponding author.

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
