# Peer review of "Experimental Investigation and Modeling of Damage Accumulation of EN-AW 2024 Aluminum Alloy under Creep Condition at Elevated Temperature"

_materials, 2021, doi:10.3390/ma14020404_

Round 1

Reviewer 1 Report

  1. From Figure 4, looks like there is no necking at all for the 100C failed specimen, but it showed ~19% of ep2. Could you explain why there is no area of reduction even with 20% strain?
  2. There is definitely dimples on the fracture surface for 100C specimen. But there are regions showing brittle failure too, right?
  3. From figure 12, can you tell where is the crack initiated, central or external region?
  4. The quality of Figure 8 is not as good as Figure 12.
  5. This is a well written paper. The authors have done a great job.

Reviewer 2 Report

Overall, the paper is well written and tells a complete story with mechanical characterisation, microscopy and modelling. It is worth publishing however I have some issues with the manuscript and model which I have listed below. Some comments would be interesting to ask, however may be beyond the scope of the current work. I have marked these with (optional).

  1. Line 74 – what do you mean by observation and registration density?
  2. Problems with explicit time dependencies such as equation 6 is they are difficult to apply to complex loading and capture any temperature, strain, or strain rate dependencies.
  3. Line 100 – what is the proposed approach? Hu et al?
  4. It is worth mentioning the work of Bryan Dyson who developed a creep damage mechanics framework, deriving physically based mathematical forms to capture the impact of different mechanisms. Dyson, B. (2000). "Use of CDM in materials modelling and component creep life prediction." Journal of Pressure Vessel Technology-Transactions of the Asme 122(3): 281-296.
  5. Paragraph 123 – you mention two factors influencing damage, but only the detail of one. Both should be described, justifying the portioning of this behaviour.
  6. Line 136 – please include (wt.%) to confirm the composition is in weight percent rather than atomic percent.
  7. Line 207 – just “twenty” rather than “twenty-some”
  8. Line 212 – what is changing in the microstructure?
  9. Table 2 – explain ep1, ep2, and the highlighted rows within the caption.
  10. A concern of the modelling approach is that the strain rate sensitivity of the tensile deformation is not accounted for. The authors need to compare the strain rate of the tensile test with those during the loading regime of the creep tests. Are the strain rates in the tensile tests comparable to the strain rates during the creep tests? If not, the initial plastic strain and the first component of the damage accumulation may not be captured accurately.
    1. What strain rates occur during the loading in the creep tests for the 100C condition?
    2. What strain rate were the tensile tests performed at?
    3. (optional) How well does Equation 9 capture the strain rate sensitivity of the alloy?
  11. The paper would benefit from an attempt to justify the treatment of damage to physical mechanisms. You should state that you assume that under creep conditions there is no work hardening behaviour in the paragraph starting on line 335 and that this captures the non-linearity of the damage accumulation at 100C presented in Figure 13. Is work hardening behaviour the only difference? A justification\discussion for this assumption is needed (what is causing work hardening and why it does not occur during creep loading).
  12. The paper needs to include model predictions for the low stress cases highlighted in grey in table 3 and include more discussion about why the model behaviour does not capture this regime. The paper is missing a figure comparing all data and model predictions. What do you think the model is missing to capture this behaviour?
  13. The model parameters have been calibrated for each temperature individually.
    1. Can you interpolate safely between these temperatures?
    2. How well does the model capture the transition from 100C to 200C where the material does not deform so much during loading?
    3. What about extrapolation?
    4. (optional) Or more complex loading conditions – transient temperature / transient loading / transient temperature and loading?
  14. (optional) The insights described in paragraph 368 are interesting. Can you include a figure to illustrate this behaviour?

Reviewer 3 Report

Dear Editor: I would like to express my deep thanks for inviting me to review the manuscript ID: materials-1076654

Title:   Experimental investigation and modeling of damage accumulation of EN-AW 2024 aluminum alloy under creep condition at elevated temperature

Authors: Adam Tomczyk, Andrzej Seweryn

Comments:

Abstract:

Please write the samples composition of the 2024 alloy.

Introduction part:

Please write the aim and novelty in this work at the end of introduction section.

Materials and methods:

Please clearly describe the test procedure.

Please include the number tensile test specimens

Results and discussion:

  1. Please provide high resolution images in Figure 7 and combined with Figure 8
  2. Please combine the Figure 9 and Figure 10.
  3. Please combine the Figure 11 and Figure 12

Conclusions

Please rewrite the conclusion section without figure

RECOMMENDATION

After reviewing the enclosed manuscript for “Materials”, the present manuscript contains some kinds of scientific analysis but it is mandatory required to modify according to the preceding remarks. So, the manuscript can be accepted for publication after minor revisions have been made.
